# EFFECTIVE PATH: KNOW THE UNKNOWNS OF NEURAL NETWORK

## ABSTRACT

Despite their enormous success, there is still no solid understanding of deep neural network's working mechanism. As such, researchers have demonstrated DNNs are vulnerable to small input perturbation, i.e., adversarial attacks. This work proposes the effective path as a new approach to exploring DNNs' internal organization. The effective path is an ensemble of synapses and neurons, which is reconstructed from a trained DNN using our activation-based backward algorithm. The per-image effective path can be aggregated to the class-level effective path, through which we observe that adversarial images activate effective path different from normal images. We propose an effective path similarity-based method to detect adversarial images and demonstrate its high accuracy and broad applicability.

## 1 INTRODUCTION

Deep learning (DL) has revolutionized the key application domains such computer vision (Krizhevsky et al., 2012a), natural-language processing (Sutskever et al., 2014), and automatic speech recognition (Abdel-Hamid et al., 2014). DL models have outperformed traditional machine learning approaches and even outperformed human beings. Although most of the current research efforts have been in improving the efficiency and accuracy of DL models, interpretability has recently become an increasingly important topic. This is because many DL-enabled or DL-based systems are mission-critical systems, such as ADAS (Dynov, 2016) and online banking systems (Fiore et al., 2017). However, to date, there is no theoretical understanding of how DL models work, which is a significant roadblock in pushing DL into mission-critical systems.

Owing to the lack of interpretability, DL models usually do not have a clear decision boundary and are vulnerable to the input perturbation. Researches have recently been proposed (Pei et al., 2017; Moosavi-Dezfooli et al., 2015; Kurakin et al., 2016a; Carlini & Wagner, 2016), which can all successfully find a small perturbation on the input image to fool the DNN based classifier. There is also prior work that demonstrates the physical attack feasibility by putting a printed image in front of a stop sign to mislead a real DNN based traffic sign detector (Eykholt et al., 2018). Last but not the least, a DNN model often fails for inputs that are dramatically different from the training samples. For example, the classification model used in Tesla's autopilot system that incorrectly classified a white truck to cloud (Golson, 2016) and caused the crash accident.

To address the vulnerability challenge in DL models, this work proposes the *effective path* as a new approach to explore the internal organization of neural networks. The effective path for an image is a critical set of synapses and neurons that together lead to the final predicted class. The concept is similar to the execution path of a control-flow based program (Ball & Larus, 1996). We propose an activation based back-propagation algorithm to extract the image's effective path, which preserves the critical information in the neural network and allows us to analyze the inner structures of DNNs.

The derived per-image effective path has the direct aggregation capability. For example, we get per-class effective path by aggregating the effective path from all training images in the same class. We can then decompose the entire DNN into multiple components, each pertaining to an inference class. We perform similarity analysis and find the phenomenon called *path specialization* that different classes activate distinctive portions of the neural network in the inference task. On the basis of the observation, we analyze the path similarity between normal and adversarial images, we uncover that when an adversarial image successfully alters the prediction result by small perturbation, the network

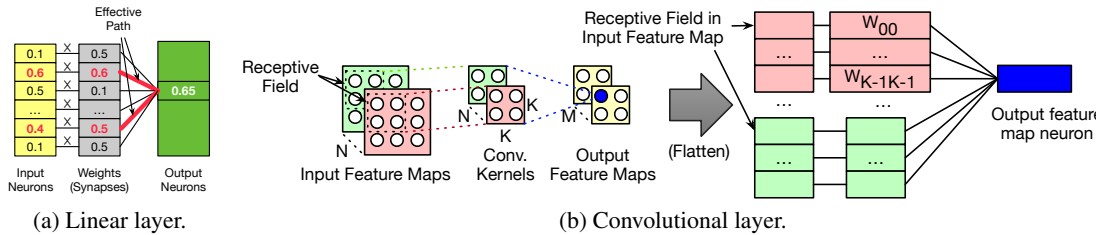

|  |  |
|---|---|
| (a) Linear layer. | (b) Convolutional layer. |

Figure 1: Examples of extracting effective paths.

activates a significantly distinctive set of effective path compared to the training samples, which lays the foundation for defending the DNN using the effective path.

For adversarial image detection, we derive a metric that is a simple linear combination of an image's per-layer effective path similarity. We perform a comprehensive evaluation on various DNN models and datasets, and show the metric can accurately detect adversarial images from six representative attack methods, for which we achieve an area under the curve value of 0.95 on MNIST and 0.89 on ImageNet. Our linear model based method offers high interpretability and has high transferability to unseen attacks. For example, we train the detection model on a single attack method, which almost provides the same detection accuracy when training the model on all attack methods. In the end, we show that effective path can not only be used for adversarial image detection but also for explaining the impact of the training process and network structure on the DNN's inference capability.

## 2 EFFECTIVE PATH

This work introduces ***effective path*** to facilitate the dissection of the black-boxed neural network. We borrow the concept from the path profiling of program analysis (Ball & Larus, 1996): a program is represented in the form of control flow graph, where a node is a basic block and an edge is the control flow between basic blocks. Compilers use path profiling to identify the sequences of frequently executed basic blocks, i.e., execution paths in the program. A program's path profile provides useful insights on its execution and facilitates the understanding of the program. Similarly, we treat a neural network as a dataflow graph where a node is a neuron and an edge is a synapse (weight) between two neurons. In the high level, the effective path represents the large dataflow inside a DNN.

**Single Image Extraction** We first explain how to extract the effective path for a single image, denoted as $\mathcal{P} = (\mathcal{N}, \mathcal{S}, \mathcal{W})$, which represents the collection of critical neurons $\mathcal{N}$, synapses $\mathcal{S}$ and weights $\mathcal{W}$. It can be further brokendown to the per-layer form $\mathcal{N} = (\mathcal{N}^1, \ldots, \mathcal{N}^L)$, $\mathcal{S} = (\mathcal{S}^1, \ldots, \mathcal{S}^L)$, $\mathcal{W} = (\mathcal{W}^1, \ldots, \mathcal{W}^L)$, where $\mathcal{N}^l$ represents the important output neurons of layer $l$, while $\mathcal{S}^l$ and $\mathcal{W}^l$ represent important synapses and weights of layer $l$.

The extraction process starts at the last layer $L$ and moves backward to the first layer. In the last layer $L$, only the neuron corresponding to the predicted class $n_p^L$ is active and thus is included in the effective path, i.e., $\mathcal{N}^L = \{n_p^L\}$. The important weights form the minimum set of weights that can contribute more than $\theta$ ratio of the output neuron $n_p^L$. Equation 1 formalizes the process, where $\tilde{K}_p^L$ is a selected set of weight indices with neuron $n_p^L$ as the output, $w_{k,p}^L$ is the weight value, and $n_k^{L-1}$ is the corresponding input neuron value (also the output neuron of layer $l-1$). To find the minimum $\tilde{K}_p^L$, we can rank the weight and input neuron pairs by the value of their product and choose the minimum number of pairs that contribute to more than threshold $\theta \times n_p^L$.

$$\min_{\tilde{K}_p^L} |\tilde{K}_p^L|, s.t. \sum_{k \in \tilde{K}_p^L} n_k^{L-1} \times w_{k,p}^L \geq \theta \times n_p^L \tag{1}$$

$$\mathcal{W}^L = \{w_{k,p}^L | k \in \tilde{K}_p^L\} \tag{2}$$

$$\mathcal{N}^{L-1} = \{n_k^{L-1} | k \in \tilde{K}_p^L\} \tag{3}$$

After deriving the weight indices set $\tilde{K}_p^L$, we can get the $\mathcal{W}^L$ set using Equation 2. Since the last layer is the fully connected layer and there is a one-to-one mapping between weight and synapses, $\mathcal{S}^L$ can also be derived. Meanwhile, since the output neurons of layer $L-1$ are the input neurons of layer $L$, it is straightforward to derive $\mathcal{N}^{L-1}$ in Equation 3. We then can repeat the process in Equation 1 for every active neuron in $\mathcal{N}^{L-1}$: each active neuron will result in a set of weights and their union form the $\mathcal{W}^{L-1}$. The process repeats backward until the first layer, and we get the whole neuron set $\mathcal{N}$, synapse set $\mathcal{S}$, and weight set $\mathcal{W}$ for the single input image.

Note that the above process applies to the fully connected layer. We transform a convolutional layer into a fully connected layer to unify the process (Fig. 1b). There are two caveats on handling the convolutional layer. First, solving of Equation 1 does not require the ranking of all input neurons but only the neurons in the receptive field of the output neuron. Second, there is no one-to-one mapping between synapse and weight because of weight sharing. As a result, multiple synapses can have the same active weights in the effective path.

**Aggregation of Training Set Images**  The direct aggregation capability of different images' effective path is a salient feature of our work compared to the prior work critical routing path (Wang et al., 2018). Our extraction process does not alter the values of the trained neural network while the prior work requires retraining for every image. By aggregating effective paths from an image group, we can obtain a larger effective path that provides higher level perspective of the whole group. Aggregating the effective path of two images $\mathcal{P}(i)$ and $\mathcal{P}(j)$ is essentially taking the union of $\mathcal{N}$, $\mathcal{S}$ and $\mathcal{W}$ on each layer, represented by $\mathcal{P}(i) \cup \mathcal{P}(j) = (\mathcal{N}(i) \cup \mathcal{N}(j), \mathcal{S}(i) \cup \mathcal{S}(j), \mathcal{W}(i) \cup \mathcal{W}(j))$, where $\mathcal{N}(i) \cup \mathcal{N}(j) = (\mathcal{N}^1(i) \cup \mathcal{N}^1(j), \ldots, \mathcal{N}^L(i) \cup \mathcal{N}^L(j))$ ($\mathcal{N}$ and $\mathcal{W}$ are similar).

In this work, we use two types of aggregated effective path for the neural network interpretation and defense. For the class-level perspective, we aggregate images from the class $c$, denoted by $\tilde{\mathcal{X}}_c$, to get the **per-class effective path** $\tilde{\mathcal{P}}_c = \bigcup_{x \in \tilde{\mathcal{X}}_c} \mathcal{P}(x)$; for the network-level perspective, we aggregate images from the whole training set $\tilde{\mathcal{X}}$ to get the **overall effective path** $\tilde{\mathcal{P}} = \bigcup_{x \in \tilde{\mathcal{X}}} \mathcal{P}(x)$.

**Effective Path Density**  The derived overall effective path is highly sparse compared to the original network. We define the weight (synapse) density of the effective path $\mathcal{D}_{\mathcal{W}}$ ($\mathcal{D}_{\mathcal{S}}$) as the ratio of weights (synapses) in the effective path over the entire weights (synapses). They can be calculated in Equation 4, where $\mathbb{W}^l$ and $\tilde{\mathcal{W}}^l$ ($\mathbb{S}^l$ and $\tilde{\mathcal{S}}^l$) is the layer $l$'s entire weight (synapse) set and weight (synapse) set in overall effective path, respectively.

$$\mathcal{D}_{\mathcal{W}} = \frac{\sum_{l=1}^{L} \left| \tilde{\mathcal{W}}^l \right|}{\sum_{l=1}^{L} \left| \mathbb{W}^l \right|}, \mathcal{D}_{\mathcal{S}} = \frac{\sum_{l=1}^{L} \left| \tilde{\mathcal{S}}^l \right|}{\sum_{l=1}^{L} \left| \mathbb{S}^l \right|} \tag{4}$$

We extracted the overall effective path for popular DNN models including LeNet-5 (LeCun et al., 1989), AlexNet (Krizhevsky et al., 2012b), ResNet-50 (He et al., 2015), Inception-v4 (Szegedy et al., 2016), and VGG-16 (Simonyan & Zisserman, 2014). Their density is 13.8%, 20.5%, 22.2%, 41.7%, 17.2%, respectively. Prior work has shown that about 80% of weights can be removed without affecting the prediction accuracy (Han et al., 2016). Our result conforms with the number, which means that extracted path preserves the information in the neural network.

The advantage of the effective path over the DNN compression is the ability to dissect the network to different components corresponding to different inference classes. As defined above, we can aggregate paths from images of the same class to get the per-class effective path. The per-class path dissects the network to different components and can be used to understand why the neural network can distinguish different classes and study the impact of changing the network structure.

## 3 PATH SIMILARITY ANALYSIS FOR NEURAL NETWORKS

After deriving the per-class effective path, we perform the path similarity analysis among different classes, which leads to a finding called **path specialization**. Different classes activate not only sparse but also a distinctive set of neurons and synapses for the inference task. This finding further motivates us to explore the distribution of effective path for adversarial examples. Specifically, we study the adversarial example in the computer vision domain, where it is in the form of small perturbation to the input image. The perturbation is small and imperceptible by human beings but can lead to an

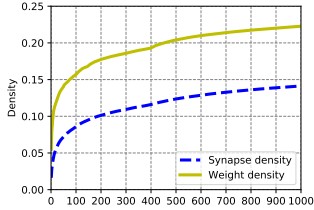 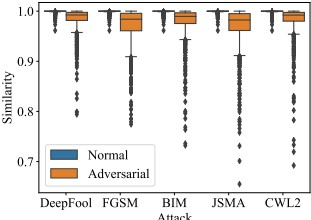

Figure 2: Class-wise path similarity in LeNet.

Figure 3: Density growth when merging per-class effective path.

Figure 4: Path similarity for normal and adversarial examples.

incorrect prediction of the neural network. Our critical insight is that adversarial images activate distinctive effective path to fool the DNN, which lays the foundation for our novel effective path based defense mechanism in Sec. 4.

### 3.1 PATH SPECIALIZATION

We first study the similarity of per-class effective paths. The similarity between class $c_1$ and $c_2$ is calculated by the Jaccard coefficient of their synapse set, i.e. $J_{c_1,c_2} = J(\tilde{\mathcal{S}}_{c_1}, \tilde{\mathcal{S}}_{c_2}) = \frac{|\tilde{\mathcal{S}}_{c_1} \cap \tilde{\mathcal{S}}_{c_2}|}{|\tilde{\mathcal{S}}_{c_1} \cup \tilde{\mathcal{S}}_{c_2}|}$.

Fig. 2 shows the class-wise path similarity in LeNet, unveiling the existence of path specialization: the averaged similarity between two classes is low (around 0.5). On average, two classes activate about 50% common paths, as well as 50% distinctive paths. Moreover, the degree of path specialization reflects the similarity between the two classes. For example, in Fig. 2, digit '1' has the highest degree of specialization (i.e., lowest path similarity against other digits) because of its unique shape. Digit '5' and '8' have the highest path similarity of 0.6 owing to their similar shape.

We observe the existence of the path specialization in other datasets and networks. Fig. 3 shows the path density growth when merging per-class (ImageNet) paths for ResNet-50. The growth of both weight and synapse follow the same trend (weight density is greater owing to the weight sharing). The density increases rapidly initially, indicating the high degree of path specialization. After 50 classes, the density still increases but at a much slower pace. This matches the class hierarchy in the ImageNet dataset, which has around 100 basic categories: different categories have a larger degree of path specialization while classes in the same categories have a smaller specialization degree.

To summarize, there exists path specialization phenomenon in trained DNNs, which we infer, offers the neural network the ability to distinguish different classes. In the next, we study how neural networks distinguish adversarial examples from normal ones with example-class path similarity.

### 3.2 SIMILARITY ANALYSIS FOR ADVERSARIAL IMAGES

On the basis of path specialization, we study the similarity of the effective path between normal images and adversarial images. For generality, we evaluate adversarial images generated by FGSM, DeepFool, BIM, JSMA, and C&W $l_2$ attacks, whose detailed introduction is explained in Sec. 4.1.

We introduce another similarity metric called ***image-class path similarity***, which indicates how many synapses in the image's effective path come from the predicted class's effective path. It can be calculated as $J_{\mathcal{P}} = J(\mathcal{S}, \mathcal{S} \cap \tilde{\mathcal{S}}_p) = |\mathcal{S} \cap \tilde{\mathcal{S}}_p|/|\mathcal{S}|$, where $p$ is the image's predicted class, $\mathcal{S}$ is the synapse set of image effective path, and $\tilde{\mathcal{S}}_p$ is the synapse set of class $p$'s effective path. Because the per-class effective path is far larger than the image's effective path, their Jaccard coefficient will be nearly zero. As such, the image-class path similarity is essentially the Jaccard coefficient between the image's effective path and the intersection set of effective path between the image and predicted class.

Fig. 4 shows the distribution of image-class path similarity for both normal images and a rich set of adversarial images in MNIST for LeNet. The similarity values for normal images are almost all 1, and note that they are not used in the training and per-class path extraction. In contrast, the similarity values for adversarial images are mostly smaller than 1, indicating effective path as a great metric to distinguish between normal and adversarial images.

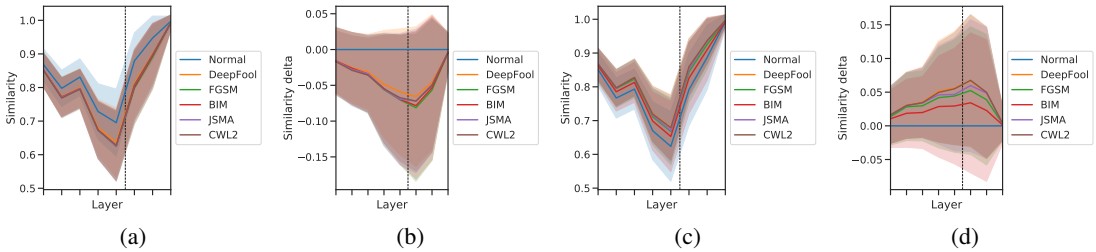

Figure 5: Distribution of per-layer similarity for AlexNet on ImageNet. Each line plot represents the mean of each kind of adversarial examples' similarity, with the same-color band around to show the standard deviation. The dashed line split convolutional layers and FC layers. (a): Rank-1 similarity. (b): Rank-1 similarity delta. (c): Rank-2 similarity. (d): Rank-2 similarity delta.

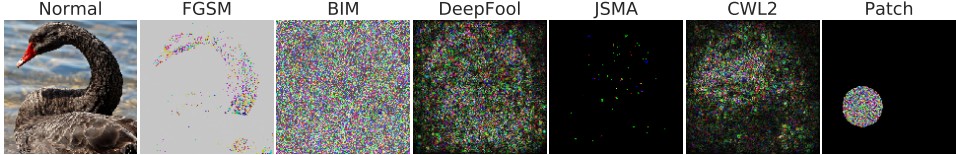

Figure 6: Normal example and perturbations from different attacks. The perturbations are enhanced by 100 times to highlight the differences.

For deeper and more complicated DNNs, we breakdown the image-class path similarity metric to different layers. It can be calculated as $J_{\mathcal{P}}^l = |\mathcal{S}^l \cap \tilde{\mathcal{S}}_p^l|/|\mathcal{S}^l|$ for layer $l$. Fig. 5a shows the per-layer similarity distribution for normal images from test set and adversarial images on AlexNet with ImageNet as training set. We find that normal images demonstrate a higher similarity degree for adversarial images. We further calculate the similarity delta, which equals the similarity value of a normal image minus the similarity value of its corresponding adversarial image. As Fig. 5b shows, all adversarial attacks cause a similarity decrease, especially in the middle layers.

Recall that we extract the effective path starting from the predicted class, i.e. rank-1 class. We also study the characteristics of effective path which starts from the rank-2 class. Fig. 5c and Fig. 5d plot the rank-2 effective path similarity for normal and adversarial images. Different from the rank-1 effective path, we find that adversarial images demonstrate a higher similarity degree for normal images. The reason is that rank-2 class in the adversarial images is typically the rank-1 class of its normal image, which still demonstrates a greater similarity than the normal images' rank-2 class.

In summary, extending class-wise path similarity to the image-class case opens the door to discover effective path's ability to detect adversarial images. The reason is that mainstream adversarial attacks modify the normally inactive path to fool the DNN. Following section details our defense mechanism.

# 4 DEFENDING NEURAL NETWORKS AGAINST ADVERSARY IMAGES

We take advantage of the observed deviation of adversarial images' similarity distribution from normal case to propose jointed similarity as a concise detection metric. This metric is generally applicable and achieves great defense performance on a wide range of different attacks, datasets, and models. Our defense mechanism is also orthogonal to other defense methods as it does no modify inputs, network structures, and training process.

**Defense Metric** Based on the per-layer similarity analysis of adversarial images, we propose *jointed similarity* as the defense metric. It can be calculated as $\tilde{J}_{\mathcal{P}} = \sum_{l=1}^{L} \omega^l J_{\mathcal{P}}^l - \sum_{l=1}^{L} \omega^{l'} J_{\mathcal{P}}^{l}{}',$ where $J_{\mathcal{P}}^l$ and $J_{\mathcal{P}}^{l}{}'$ are respectively rank-1 and rank-2 similarity for layer $l$, $\omega^l$ and $\omega^{l'}$ are their coefficients that satisfy $\omega^l \geq 0, \omega^{l'} \geq 0$. The joint similarity reflects the low rank-1 similarity degree and high rank-2 similarity degree of adversarial images. An image is detected as adversarial image if its joint similarity is less than a threshold. The joint similarity avoids overfitting and offers strong interpretation ability as it is a low-dimension linear model.

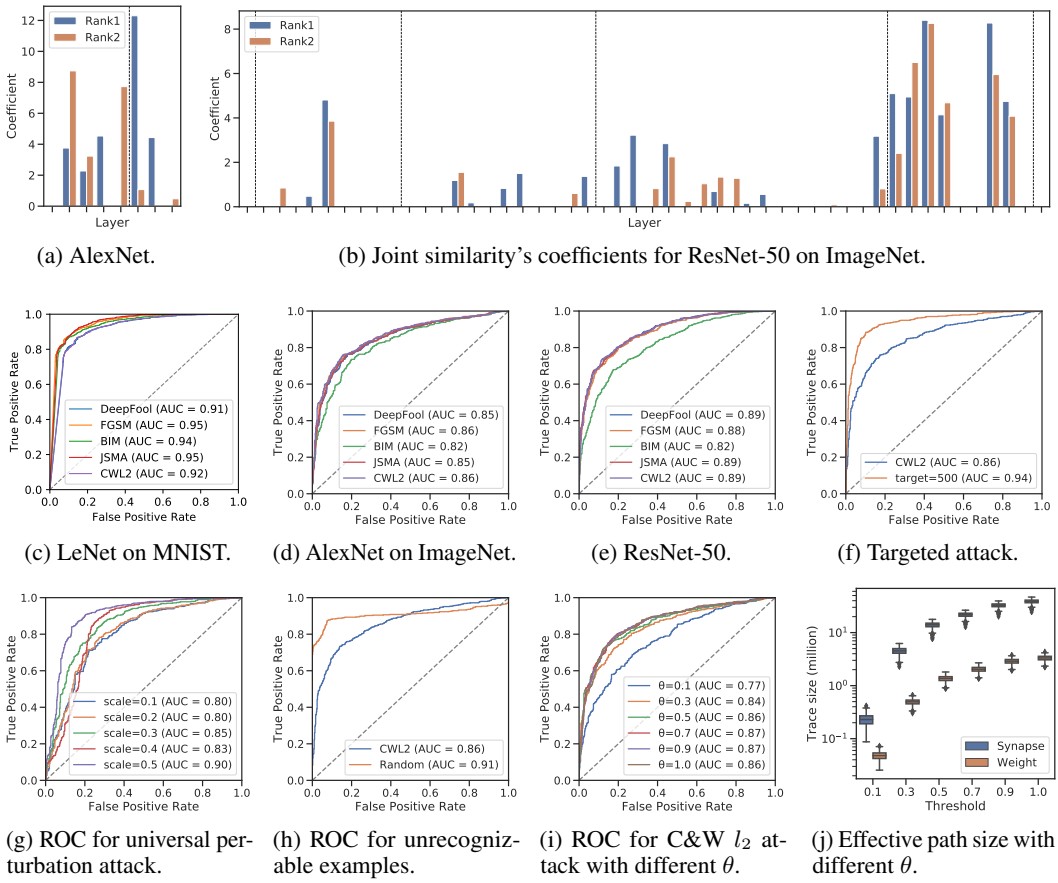

Figure 7: Detection results for LeNet, AlexNet, and ResNet-50 with joint similarity.

## 4.1 EVALUATION

To obtain joint similarity's coefficients, we train it with dataset containing both normal and adversarial examples. We use a set of representative to evaluate its detection accuracy and generality.

**Attack Approaches** We use 6 different attacks for evaluation as shown in Fig. 6. For each attack, we always choose the canonical implementation. We use Foolbox (Rauber et al., 2017) implementations and its default parameters in version 1.3.2 for Fast Gradient Sign Method(FGSM) (Goodfellow et al., 2014), Basic Iterative Method(BIM) (Kurakin et al., 2016a), DeepFool (Moosavi-Dezfooli et al., 2015), Jacobian-based Saliency Map Attack(JSMA) (Papernot et al., 2015). For Carlini and Wagner(C&W) attacks (Carlini & Wagner, 2016), we use the open-source code released by the paper authors. We use adversarial patch (Brown et al., 2017) implementation provided in CleverHans (Papernot et al., 2018) and extend it to support AlexNet without modification to its settings.

**Coefficients Training** We use LeNet-5 on MNIST, AlexNet on ImageNet and ResNet-50 v2 on ImageNet for evaluation. For each dataset, effective path extraction is performed on the overall training set with $\theta = 0.5$. For each model, adversarial examples from all evaluated attacks are aggregated and split into 80% for the training of joint similarity's coefficients and 20% for defense performance evaluation. Notice that we only generate adversarial examples for the first test image in each class of ImageNet due to the significant computational cost. The training of joint similarity's coefficients is performed by SGD running 10000 epochs, with elastic-net regularization ($l_1$ ratio is 0.5) for sparsity. Fig. 7a and Fig. 7b show the coefficients's distribution: large coefficients locate near the FC layer, which match the results from our previous analysis results.

**Against Non-targeted Attacks** Non-targeted attacks are free to use any class as the adversarial image's inference result. We evaluate non-targeted attacks with three different norms: FGSM and

Table 1: Comparison with other defenses.

| Type | Defense | $l_0$ | $l_2$ | $l_\infty$ | Generality | Transferability | Scale |
|---|---|---|---|---|---|---|---|
| | **Effective Path (This Work)** | **Y** | **Y** | **Y** | **all discussed** | **strong** | **ImageNet** |
| Detector | Metzen et al. (2017) | - | **Y** | **Y** | (non-)targeted | weak | CIFAR-10 |
| | Meng & Chen (2017) | - | **Y** | **Y** | (non-)targeted | - | CIFAR-10 |
| | Wang et al. (2018) | - | - | **Y** | targeted | - | **ImageNet** |
| Adversarial | Madry et al. (2017) | - | **Y** | **Y** | (non-)targeted | weak | CIFAR-10 |
| Training | Na et al. (2017) | - | - | **Y** | (non-)targeted | weak | CIFAR-10 |
| Input | Guo et al. (2018) | - | **Y** | **Y** | (non-)targeted | - | **ImageNet** |
| Transformation | Buckman et al. (2018) | - | - | **Y** | (non-)targeted | - | CIFAR-100 |
| Randomization | Xie et al. (2018) | - | **Y** | **Y** | (non-)targeted | - | **ImageNet** |
| | Dhillon et al. (2018) | - | - | **Y** | (non-)targeted | - | CIFAR-10 |
| Generative | Schott et al. (2018) | **Y** | **Y** | **Y** | (non-)targeted + unrecognizable | **strong** | MNIST |
| Model | Samangouei et al. (2018) | - | **Y** | **Y** | (non-)targeted | - | MNIST |

BIM with $l_\infty$ norm, DeepFool and C&W $l2$(CWL2) attack with $l_2$ norm, and JSMA with $l_0$ norm. For LeNet, we achieves an area under the curve (AUC) value up to 0.95 in Fig. 7c. Even the lowest AUC value is 0.92, because of significant path similarity distinction between adversarial and normal images of MNIST. On ImageNet, we achieve AUC of 0.85~0.86 for AlexNet and AUC of 0.88~0.89 for ResNet-50, which has more layers to provide richer information for detection, leading to the better accuracy. The BIM has a low AUC value of 0.82. The reason is that BIM iteratively modifies all pixels (Fig. 6), which makes its rank-2 effective path behave slightly different from other attacks.

**Against Targeted Attack**  Targeted attacks are designed to mislead the prediction to a specific target class. Fig. 7f shows the result of evaluating targeted C&W $l_2$ attack for AlexNet. We achieve AUC of 0.94, which is better than the non-targeted version. It is reasonable since targeted attack's stricter constraint for target class requires larger perturbation, which eases our detection.

**Against Universal Perturbation Attack**  Universal perturbation attacks generate perturbations that fool models on a large range of examples. Adversarial Patch (Brown et al., 2017) is an attack that generates universal perturbations in form of image patches, which is robust against patch transformations such as translations, rotations or scaling. The result of adversarial patches in Fig. 7g indicates that the detection becomes more accurate when the patch becomes larger. Our method can reach AUC of 0.9 when patch scale relative to image size rises to 0.5.

**Against Unrecognizable Examples**  Adversarial examples are usually human-recognizable, however, unrecognizable images can also fool neural networks (Nguyen et al., 2015). We find that effective path can also be used to detect unrecognizable examples by evaluated on LeNet and AlexNet. For LeNet, our detector can recognize 93.85% randomly generated images. For AlexNet, our method achieves AUC of 0.91 as shown in Fig. 7h. In this sense, effective path offers the DNNs the ability to identify its recognizable inputs' distribution.

**Parameter Sensitivity**  The single tunable parameter of effective path extraction is $\theta$. We test C&W $l_2$ attack under $\theta$s varying from 0.1 to 1.0 in Fig. 7i. The detection performance remains almost unchanged when $\theta$ is in range of 0.5 and 1.0, and decreases from $\theta = 0.3$. It indicates that our method is not sensitive to $\theta$. Fig. 7j shows that the effective path size under $\theta = 0.3$ decrease by one order of magnitude compared with $\theta = 1.0$, with slightly lower detection accuracy. We choose $\theta = 0.5$ as default value to save storage space and improve extraction performance without accuracy loss.

**Transferability**  Transferability measures a defense's ability to withstand unknown attacks. To figure out joint similarity's transferability, we train its coefficients for targeted and non-targeted C&W $l_2$ attacks and then test on other attacks. For non-targeted attacks, AUCs of FGSM/BIM/DeepFool/JSMA separately increase 0.01~0.02, which indicates that our method is perfectly transferable to the same type of attacks. When transfer to other types, AUC decreases 0.06 for adversarial patch and 0.03 for unrecognizable examples.

To summarize, our effective path based detection method achieves high detection accuracy without requiring attack-specific knowledge. Meanwhile, it also offers strong transferability to unseen attacks.

## 4.2 COMPARISON WITH OTHER DEFENSES

To compare our defense method with prior work, we first categorize various defenses methods to the five types listed in Tbl. 1. Since almost all the compared work reported a similar detection accuracy

(AUC value 0.9 ~0.95), we focus the comparison on the comprehensiveness, generality, transferability, and scale of their evaluation. The "-" in the table indicates that there are not enough details or experimental results to deduce an appropriate conclusion.

**Detector**    Our work fits in the detector category, which does not require any modification to inputs, models, or training process. Metzen et al. (2017) trains a DNN from network activations to detect adversarial examples. The detector subnetwork doesn't generalize well across different attack parameters or attack types because the activation values are highly attack-specific, which motivates Meng & Chen (2017) to propose MagNet. MagNet uses a reformer to move adversarial examples to normal examples' manifold. However, Carlini & Wagner (2017) shows that MagNet can be defeated by a little increase of perturbation.

The closest work to ours is Wang et al. (2018), which uses the importance coefficients of different channels in the network (named critical data routing paths, abbr. CDRPs) to detect adversarial examples. However, CDRPs do not have aggregation capability as a single channel can have different significance values for different images. As such, CDRPs fail to defend non-targeted attacks and have weak transferability. In comparison, we use the effective path, which is essentially a binary value for each neuron/synapse, and therefore can be directly aggregated. Our method generalizes well for different attacks and provides strong transferability.

**Adversarial Training**    Adversarial training requires additional training step to protect the DNNs. It has two known disadvantages: it is difficult to perform in the large-scale dataset like ImageNet (Kurakin et al., 2016b), at the same time easy to overfit to the trained kinds of adversarial examples. Even adversarial training proposed by Madry et al. (2017), considered as the only effective defense among white-box-secure defenses at ICLR 2018 (Athalye et al., 2018), is found overfitting on the trained $l_\infty$ metric (Sharma & Chen, 2017).

**Input Transformation**    Input transformation requires modification of input examples. Many image transformations like rescaling, bit-depth reduction and compression can disturb attacks and increase the perturbation lower bound, with the sacrifice of classification accuracy. This kind of defense method works less well for patch-based attacks and does not provide the ability to filter unrecognizable examples.

**Randomization**    Randomization defenses apply random modifications to model weights. They can increase required distortion by forcing attacks to generate transferable adversarial examples over a series of possible modified models. However, they also stochastically alter the prediction results, leading to the overhead of more forward passes or retraining steps.

**Generative Model**    Generative model based defenses change the classification model. They project the inputs onto the manifold before classification. Schott et al. (2018) propose a classification model that shows good generality and transferability on MNIST, but its performance on large dataset like ImageNet is still obscure. GAN-based defenses are also hard to apply in ImageNet scale due to its computational cost.

## 5    CONCLUSION AND FUTURE WORK

This work has shown the feasibility of decomposing a deep neural network into different functional blocks corresponding to different inference classes, similar to the various functional units of the brain cortex. The functional block is called effective path and constructed through back-propagation using training images. Through analysis, we find that adversarial images can activate functional blocks different from normal images to fool the DNN's prediction results because all the blocks are connected. We propose a defense method that only uses the information from the training set and the image itself, without requiring any knowledge of a specific attack. The defense method achieves high accuracy and broad coverage of mainstream attacks.

Besides adversarial defense, the effective path can also be used to understand the DNN's working mechanism. In the appendix, we report our preliminary result on how the training process and different DNN topology affects the effective path density and similarity. We believe that the functionality based decomposition is a promising direction for understanding DNNs.

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

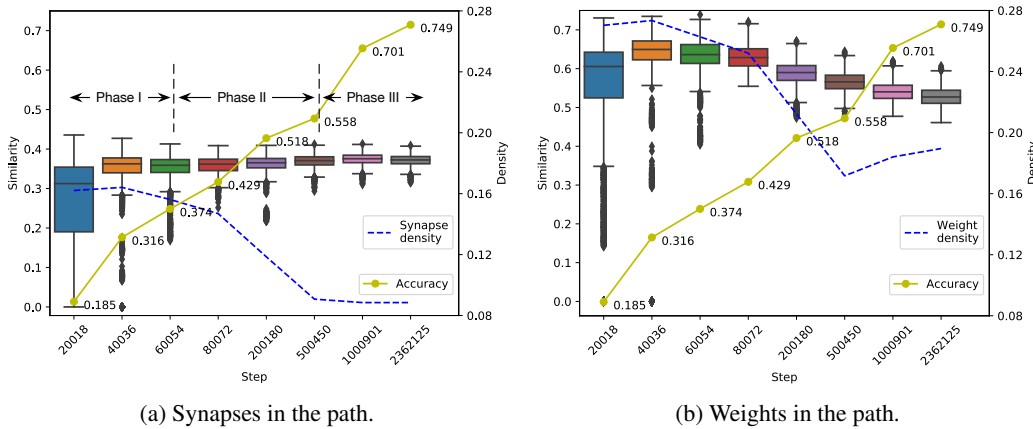

(a) Synapses in the path.

(b) Weights in the path.

Figure 8: Effective path's density and class-wise path similarity in the training process.

# APPENDIX

## A    EFFECTIVE PATH EXTRACTION FOR MORE NETWORK STRUCTURES

For the sake of brevity, we only introduce effective path extraction for networks consist of convolutional layers and FC layers in Sec. 2. We further explain other common network structures' extraction methods in this section.

**Skip Connection**    To handle skip connections in ResNet, we need to merge neurons contributed from two different layers. Consider a skip connection from layer $l$ to layer $l + m$, then active neurons in layer $l$ are collected from layer $l + 1$ and $l + m$, denoted as $\mathcal{N}^l = \{n_k^l | k \in \tilde{K}^{l+1} \text{ or } k \in \tilde{K}^{l+m}\}$, where $\tilde{K}^{l+1}$ and $\tilde{K}^{l+m}$ are the selected sets of weight indices in layer $l + 1$ and $l + m$ respectively.

**Pooling Layer**    Pooling layers can be treated as the special case of convolutional layers during extracting. For average pooling layer, we treat it as a convolutional layer with all weights equal to 1; for max pooling layer, we treat it as a convolutional layer that always picks rank-1 weight and input neuron pair when finding the minimum $\tilde{K}_p^l$.

## B    FURTHER STUDY OF NEURAL NETWORK INTERPRETABILITY

We further study how the training process and different network structure impacts the path specialization. In the next, we study how the training process and different network structures impact the path specialization.

### B.1    TRAINING PROCESS

We study how the training process transforms a randomized network to the final state from the perspective of the effective path. Specifically, we extract the effective path for each class at different training stages. Through the analysis, we find that the training process contains three distinctive phases with different path's density and similarity trend, which share similar insights from the previous work using information bottleneck theory to explain training process (Wolchover, 2017).

Fig. 8 shows training process for ResNet. We choose different stages in training and show the class-wise path similarity in the form of box-plot, on top of which we also overlay the path density and prediction accuracy. In the first phase, the density of synapses and weights in the effective paths stays the same while their similarity increases. In the beginning, the network is in a randomized state and simply tries to memorize the input data.

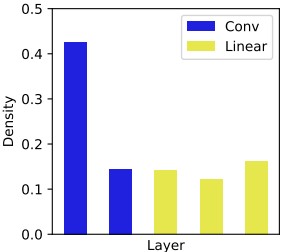 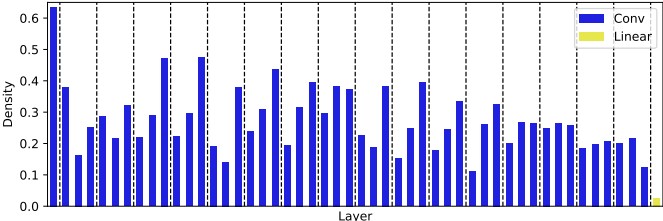

Figure 9: Per-layer density of effective paths in LeNet.

Figure 10: Per-layer density of effective paths in ResNet-50. Layers in ResNet-50 is organized into 3-layer bottleneck blocks, which is split by dashed lines.

In the second phase, the density of both synapses and weights decrease rapidly. The similarity of the synapses stays relatively the same while the similarity of weights decreases. In this phase, the network mainly performs compression, and the path specialization mainly manifests in the form of weights. In other words, the network tries to use class-specific features extracted by different convolutional filters to increase the specialization degree.

In the third phase, the synapse density stops to decrease but weight density starts to increase. Meanwhile, the weight similarity continues to decrease. In this phase, the network compression stops and mainly relies on path specialization (via weight) to increase the prediction accuracy. The path specialization even causes the weight density increases a bit.

In summary, we find that the training process contains mainly three phases, first two of which conforms to the memorization and compression phase identified by the prior work (Wolchover, 2017). The second phase performs compression (less density) and path specialization (less similarity), while the third phase mainly includes the path specialization. After these phases, the network is transformed into a state with sparse and distinctive paths with great inference capability.

## B.2 NETWORK STRUCTURE

After establishing the effective path as a great indicator of the neural network's inference performance, we study that how the network structure affects the effective path characteristics.

Fig. 9 shows the per-layer path density in LeNet. We observe the first convolutional layer has a much higher density compared to the following layers, which matches with the established knowledge that the shallow layers in a CNN extract high-level features that are shared by different classes.

CNN designers have found using a deeper network can increase the prediction accuracy to a certain degree. However, the accuracy stops to increase after a certain number of layers owing to the vanishing gradient in the training process. As such, the ResNet structure with skip connection was proposed to overcome this difficulty. Fig. 10 shows the per-layer path density for ResNet-50. Not only the first two layers still have higher density, but also layers before a skip connection also have high density. This suggests that skip connection helps not only the gradient propagation but also the effective paths formation. In the end, ResNet is able to converge and achieve great prediction performance.

## C  SUPPLEMENTAL ANALYSIS

### C.1  PER-LAYER SIMILARITY ANALYSIS FOR RESNET-50 ON IMAGENET

Per-layer similarity distribution for ResNet-50 on ImageNet is shown in Fig. 11. Similar with AlexNet, adversarial images lead to lower rank-1 similarity and higher rank-2 similarity compared with normal images. Furthermore, corresponding to AlexNet's FC layers, the largest similarity delta is also located in last several layers.

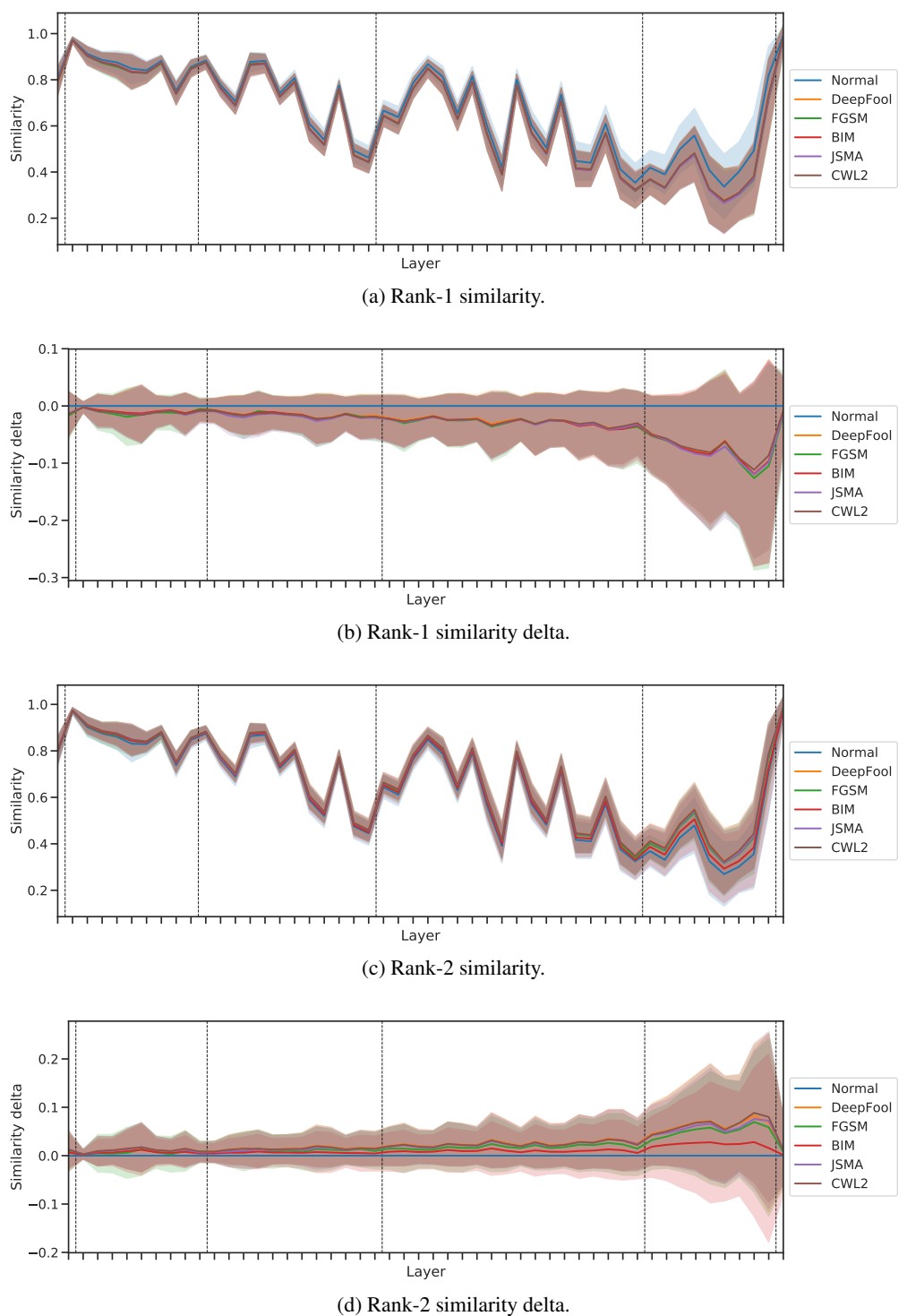

(a) Rank-1 similarity.

(b) Rank-1 similarity delta.

(c) Rank-2 similarity.

(d) Rank-2 similarity delta.

Figure 11: Distribution of per-layer similarity for ResNet-50 on ImageNet. Each line plot represents the mean of each kind of adversarial examples' similarity, with the same-color band around to show the standard deviation. The dashed lines indicate that down-sampling is performed in the next layer.

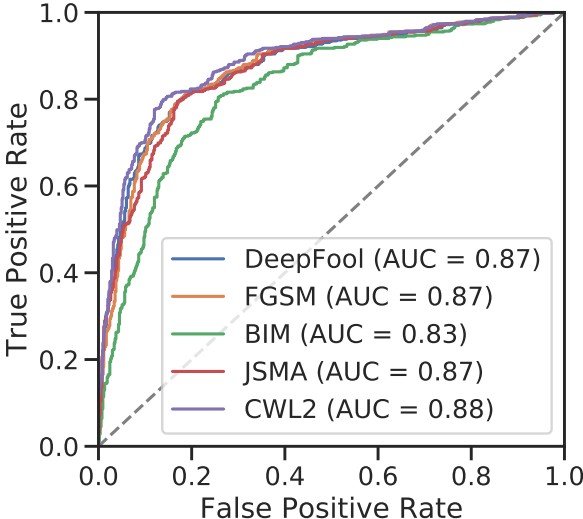

Figure 12: ROC for AlexNet on ImageNet with weight-based joint similarity.

## C.2 WEIGHT-BASED JOINT SIMILARITY AS DEFENSE METRIC

For adversarial detection, we can use information from model weights as alternative of synapses. By calculating image-class path similarity from weights in effective path instead, i.e., let $J^l_{\mathcal{P}} = |\mathcal{W}^l \cap \tilde{\mathcal{W}}^l_p|/|\mathcal{W}^l|$ for layer $l$, we obtain weight-based joint similarity. The detection result using weight-based defense metric for AlexNet is shown in Fig. 12, which indicates that it achieves as high accuracy as the synapse-based metric.

