# OpenReview forum: "Effective Path: Know the Unknowns of Neural Network"
_ICLR.cc/2019/Conference_

### Official Review · AnonReviewer1 · 2018-11-01
**Review for "Effective Path: Know the Unknowns of Neural Network"**

**Rating:** 6
**Confidence:** 5

**Review:**

This paper proposes a measure (“effective path”) of which units and weights were most important for classification of a particular input or input class. Using the effective path, the authors analyze the overlap between paths across classes for CNNs and between adversarially modified and unmodified images. Finally, the paper proposes an adversarial defense method based on effective path which detects adversarially manipulated images with high accuracy and generality to a variety of settings.

Overall, this paper is interesting and provides several novel observations. The clarity of the exposition is generally good, but can be improved in several places (mentioned below). As for significance, effective path is likely to inform future analyses of neural networks, and the adversarial defense may prove impactful, though ultimately, its impact will depend on if and when the defense is broken.

However, there are several important controls missing from the analysis, several claims which are unsubstantiated, and experimental details are lacking in a few places. As such, in its current form, I can only weakly recommend this paper for acceptance. If in the revision the controls requested below are included, additional evidence is provided for the unsubstantiated claims (or if those claims are toned down), and exposition of missing experimental details is included, I’d be happy to raise my score.

Major points:

1) While the observation regarding path specialization is very interesting, one cannot gauge whether or not the degree of overlap observed between class-specific paths signals path specialization or simply high input-to-input path variance (which is similar both within and across classes). In order to distinguish between these possibilities, a measure of intra-class path similarity is necessary. In addition, an experiment similar to that in Figure 2 with CIFAR-10 would be quite helpful in evaluating whether this phenomenon exists in more natural datasets (the ImageNet results are difficult to interpret due to the large number of classes).

2) Several claims in the path specialization section are unsubstantiated.

2a) In particular, the claim that ‘1’ has the highest degree of specialization “because of its unique shape” is made without evidence as is the similarity between ‘5’ and ‘8’. ‘6’ is also similar to ‘8’ and yet does not show the same similarity in the path specialization. These differences may very well simply be due to chance.

2b) The claim that the path specialization in ImageNet matches the class hierarchy is made only based on the rough non-linearity of Figure 3. Please either measure the overlap within and across class categories or soften this claim.

3) The similarity analysis for adversarial images is also very interesting, but a comparison between unmodified and randomly perturbed images with matched norms to the adversarially perturbed images is necessary to establish whether this effect is due to noise generally or adversarial noise.
It’s unclear how the effective path is calculated when negative weights are involved. Further exposition of this aspect would be helpful.

Minor points/typos:

1) There are several places where confusing concepts are introduced in one paragraph but explained several paragraphs later. In particular, the distinction between synapses and weights is introduced halfway through page 2 but explained on page 3 and the fact that the coefficients for the defense metric are learned is unclear until page 4 even though they’re introduced on page 3.

2) Typos:

2a) Section 1, fourth paragraph: “...and adversarial images, we uncover...” should be “...and adversarial images, and we uncover...”

2b) Section 1, fourth paragraph: “...by small perturbation, the network…” should be “...by small perturbations, the network…”

2c) Section 2, first paragraph: “...the black-boxed neural…” should be “...the black-box neural…”

2d) Section 2, first paragraph: “In the high level…” should be “At a high level…”

2e) Section 4, first paragraph: “...as it does no modify…” should be “...as it does not modify…”

2f) Title, should be "Neural Network"?

---

### Official Review · AnonReviewer2 · 2018-11-04
**No clear contribution showed without experiment comparison with previous work. Also the motivation could be more clear.**

**Rating:** 4
**Confidence:** 3

**Review:**

The authors propose the notion of effective path, for the purpose of identifying neurons that contributes to the predictions and being able to detect adversarial images in the context of image classification.
Overall the paper is well written except that the authors are mixing two highly related but still different topics: explanation and adversary detection so that the motivation is confusing.
The experimental results indeed show promises that effective path can help understand class similarities and network efficiencies but doesn’t really show how the proposed work is adding value to the field.
It lacks the experimental comparison with previous methods but only include discussion in texts.
This paper could turn out to be a stronger paper but it is not ready yet.

Below are some more detailed comments.
1) The authors motivate by stating that the vulnerability of NN to input perturbations is due to the lack of interpretability (Section Introduction & Abstract). I can understand that we want more interpretability, and we want less vulnerability, but I can’t agree that vulnerability is caused by lack of interpretability. Also, the authors are trying to accomplish both tasks, interpretability and adversary detection, by showing data analysis of how the findings coincide with prior knowledge (eg. Class of digit 1 is the most different from other classes in MNIST task), and by showing detecting adversary images. However, neither has valid quantitative comparison with previous work; actually for the interpretability topic, the authors didn’t really provide a tool or a generalizable method. Thus, I would suggest to choose one of the two topics (ie. adversarial image detection) and focus on it by adding thorough comparison with other methods; in the discussion and result section, include the interpretability analysis to justify why the proposed adversary detection method is behaving in certain ways.

2) One topic that is missing from the paper is the time complexity of the proposed method. At a naïve estimate, it would require tracking and finding the minimum set of effective neurons with threshold \theta and thus per instance, at least O(m log m) is required at prediction phase, where m is the number of features; for n instances, the asymptotic complexity is O(nm log m) How does it compare to the other adversary detection methods?

3) Page 3 mentions that the work for critical routing path (Wang et al. 2018) requires re-training for every image; this statement is not really true without more context. Also authors discuss this work again very briefly in Page 8 due to the high similarity in methods and motivation with the proposed method, but the authors don’t show any quantitative comparison. After all, both methods are trying to identify neurons that contribute the most to the prediction, some more concrete comparison would be nice.

4) Page 3 mentions that the derived overall effective path is highly sparse compared to the original network and the effective path density for five trained models ranges from 13% to 42% which conforms with the “80%” claim from another paper. Together with the other similar statements, it would be really nice to note what \theta is used for such statements; how does such statement change with different \theta. Also some discussion would be nice about what such sparsity implies. Specifically, does the sparsity suggest the opportunity for feature selection, or does it suggest a way for detecting overfitting?

5) Page 5 shows the path similarity between the normal and the adversary examples; from the figure 5a and 5b, we can see the on the first layer, the mean deviate between normal and others but why the last layer they almost reach to the same point? It seems it is the middle layer that distinguish the normal from the adversary examples the most. Some more discussion would be good.

6) Some justification of why \theta=0.5 is chosen would be good on Page 6.

7) On Page 7, the authors are discussing the performance of the proposed method, however, there is no really comparison with other methods. But rather, the authors stated “better accuracy”, “AUC… is better…” by comparing different evaluation scenarios. I don’t find such discussion helpful in showing the contribution of the proposed method. Also in the parameter sensitivity, it would be nice to add the analysis for the effective path density and see if it still conforms with the “80%” claim with different \theta.

8) Page 1, need to add citations for the statement “… and even outperformed human beings.”

9) Minor issue: Page 1 “such computer vision…” should be “such as computer vision…”.

---

### Official Review · AnonReviewer3 · 2018-11-04
**Critical paths in DNNs are different for adversarial examples, but how effectively can they be used for their detection?**

**Rating:** 4
**Confidence:** 4

**Review:**

This paper proposes a method for the detection of adversarial examples based on identification of critical paths (called "effective paths") in DNN classifiers. Borrowing from the analysis of execution paths of control-flow programs, the authors use back-propagation from the neuron associated from the final class decision to identify a minimal subset of input synapses accounting for more than a threshold proportion ("theta") of the total input weight. The identification process is then recursively applied at the preceding layer for those neurons associated with the selected minimal subset of synapses, forming a tree of synapses (the "effective path"). The authors then propose to compare the effective paths (actually, unions of paths) of different examples using simple structural dissimilarity measures, which they extend to allow comparison to a typical (aggregated) path for multiple examples drawn from a common class.

In their experimentation with their measure, they noted that examples generated by a number of adversarial attacks tend to be less similar to their first-ranked estimated class than normal examples are to their own first-ranked classes. Similarly, they note that these same adversarial attacks tend to be *more* similar to their second-ranked classes than normal examples are to their own second-ranked classes (as the authors point out, this is likely due to the increased likelihood of the second-ranked class of adversarial examples being the true class for the original example from which it was perturbed). The authors then propose the difference between these two similarities (that is, first-ranked dissimilarity minus second-ranked dissimilarity) as a characterization of adversarial examples.

The idea of using critical paths in the DNN to detect adversarial examples is interesting, and the authors deserve credit for showing that these critical paths (as defined in this paper) do show differences from those of normal examples. However, the originality of the approach is undercut by the recent work of Wang et al. (CVPR, 2018), which the authors acknowledge only in the discussion of experimental results. Although the details are different as to how critical paths are identified, and how adversarial examples can be detected using them, the strategies are definitely related - a more detailed explanation of this should have been given in the introduction of the paper. More troubling is the fact that a head-to-head experimental comparison is not provided, neither with Wang et al. nor with other state of the art detectors, other than a qualitative assessment of the capabilities of some detectors in Table 1. Note that even this qualitative discussion does not include some of the recent detection approaches, such as BPDA (Athalye et al., ICML 2018) or LID (Ma et al., ICLR 2018).

The question of how best to define critical paths and their similarities is still very much open - the authors' approach is rather simplistic and straightforward. For example, is their similarity measure biased towards the contributions from early layers? Can a layer-by-layer weighting of contributions improve the performance?

The authors do not always interpret their own experimental results correctly. For example, their results in Figures 7i and 7j don't really support their conclusion that performance "remains almost unchanged" when theta is in the range 0.5 - 1.0. Also, Figure 4 does not show that their effective path similarity is not *directly* "a great metric to distinguish between normal and adversarial" examples, because a large proportion of adversarial examples have scores that fall in the typical range for normal examples (however, there are differences in tendency which can be exploited, as the authors do show).

The organization of the paper is in some need of improvement. For example, the discussion of densities of "effective paths" (Section 2) comes well before the details of the choice of threshold value theta used to generate them (Section 4.1).

To summarize:

Pros:
* A good case is made for the use of critical paths as a way of differentiating adversarial examples from normal examples.
* The reported improvement in similarity of adversarial examples with respect to their second-ranked classes is particularly intriguing.
* The paper is generally well written and easy to follow.

Cons:
* The experimental treatment is insufficient; in particular, a more carefully considered experimental justification is needed with respect to other detection strategies.
* The question of how best to define critical paths and their similarities is still very much open.
* The authors do not always interpret their own experimental results correctly.
* The organization of the paper is in some need of improvement.

---

### Meta-Review · Area_Chair1 · 2018-12-14

**Confidence:** 4
**Recommendation:** Reject

**Metareview:**

The paper presents an approach to estimate the "effective path" of examples
in a network to reach a decision, and consider this to analyze if examples
might be adversarial. Reviewers think the paper lacks some clarity and
experiments. They point to a confusion between interpretability and adversarial
attacks, they ask questions about computational complexity, and point to some
unsubstanciated claims. Authors have not responded to reviewers. Overall, I
concur with the reviewers to reject the paper.